# Clinical Case: Patient with Mixed Graft Rejection Four Days after Kidney Transplantation Developed Specific Antibodies against Donor Bw4 Specificities

**DOI:** 10.3390/antib10030028

**Published:** 2021-07-21

**Authors:** Claudia M. Muñoz-Herrera, Juan Francisco Gutiérrez-Bautista, Miguel Ángel López-Nevot

**Affiliations:** 1Clínica Imbanaco Grupo Quirónsalud, Cali 760042, Colombia; 2Servicio de Análisis Clínicos e Inmunología, Hospital Universitario Virgen de las Nieves, 18014 Granada, Spain; manevot@ugr.es

**Keywords:** kidney rejection, donor specific antibodies, anti-Bw4, mixed rejection, Banff classification

## Abstract

Kidney transplantation, like other transplants, has the risk of producing graft rejection due to genetic differences between donor and recipient. The three known types of renal rejection are listed in the Banff classification: T-cell-mediated rejection (TCMR), antibody-mediated rejection (ABMR), and mixed rejection. The human leukocyte antigens (HLA) are highly polymorphic and may be the targets of donor-specific antibodies, resulting in ABMR. Therefore, prior to transplantation, it is necessary to analyze the HLA genotype of the donor and recipient, as well as the presence of DSA, in order to avoid hyperacute rejection. However, due to the shortage of kidneys, it is very difficult to find a donor and a recipient with completely matched HLA genotypes. This can trigger a future rejection of the kidney, as is reported in this work. We describe a patient who received a kidney transplant after a negative DSA test, who developed graft rejection with antibodies against the donor’s HLA-Bw4 public epitope and lymphocytic infiltrate four days after transplantation, whose differential diagnosis was mixed rejection.

## 1. Introduction

For patients with end-stage renal failure, kidney transplantation is the treatment of choice, since it provides a better quality of life and greater survival compared to dialysis therapy [1]. In kidney transplantation, as with the transplantation of other solid tissues, there is a risk that the transplanted organ may be rejected by the recipient. Rejection occurs when the recipient’s immune system recognizes the transplanted organ as foreign [2]. This threat of rejection is increased when patients are transplanted with organs from a deceased donor, since prolonged ischemia increases the expression of adhesion molecules, the release of cytokines and chemokines, as well as the expression of HLA (Human Leukocyte Antigen) antigens [3]. On the other hand, in transplants carried out with a living donor, the ischemia time is shorter and, therefore, a lower rejection rate has been observed [4].

Incompatibility in HLA molecules between the donor and the recipient represents one of the main barriers to transplant success. Despite the immunosuppressive, donor-specific antibodies (DSA) directed against the donor’s HLA molecules can be produced, resulting in antibody-mediated rejection (ABMR) of the transplant [5]. DSA can appear at any time after transplantation, as a consequence of insufficient immunosuppression [6].

HLA molecules are the most polymorphic molecules known, therefore, the compatibility that exists between the donor and the recipient is crucial for the long-time survival of the transplanted organ. The difference between the various HLA molecules lies mainly in the peptide-binding zone, where the variation in a few amino acids is sufficient to cause the rejection of the graft [2]. These variations of amino acids in the HLA molecules are called “private epitopes”. In addition, the HLA molecules of the B locus present two possible polymorphisms in the α1 domain, formed by residues 77, and residues 80 to 83. The resulting epitopes, which are common to several B alleles, are called Bw4 or Bw6, known as “public epitopes”. These public epitopes divide HLA-B alleles into two groups, those that express Bw4 and those that express Bw6 [7]. Other alleles that also express the Bw4 public antigen are HLA-A A23, A24, A25 and A32 [8].

There are three types of rejection in kidney transplants: T-cell-mediated rejection TCMR, ABMR, and mixed rejection [9,10]. TCMR begins with the presentation of the donor alloantigen’s to the recipient’s T cells, inducing their activation, proliferation and subsequent migration to the graft where they secrete cytokines that induce damage and injury in the transplanted organ [11]. ABMR is produced by the recognition of the donor’s HLA or non-HLA molecules; and mixed rejection, as the name implies, is the combination of cell-mediated and antibody-mediated rejection. The rejection can also be classified by the time it is presented as hyperacute (within 48 h after transplantation), acute (within the first 6 months after transplantation) or chronic (a slow and progressive reduction in kidney function, that persist for years) [12,13,14].

The antibodies formed can induce graft injury by binding to the organ’s endothelial cells, triggering the activation of the complement cascade, which leads to cell lysis and the possible activation and recruitment of cells with Fc (crystallizable fragment) receptors, such as the neutrophils and NK cells. Lesions induce platelet aggregation and cell recruitment, leading to thrombus formation, increasing the risk of graft failure [15]. The diagnosis of ABMR, according to the Banff classification [9], requires that three characteristics are met: 1—morphological changes, including microvascular inflammation characterized by infiltration of neutrophils and the presence of mononuclear cells in the glomeruli and peritubular capillaries. 2—presence of acute tubular injury, thrombotic microangiopathy or, intimal or transmural arteritis. 3—evidence of complement activation by deposition of C4d in peritubular capillaries or within arterial fibrinoid necrosis and the presence of DSA [16]. However, there are cases in the presence of DSA ABMR without evidence of C4d deposits, called rejection mediate by non-complement fixing antibodies. The ability of antibodies to activate complement depends on their affinity to bind to the C1q molecule, the first step in this lytic pathway, which also determines its cytotoxic capacity [17]. Therefore, the absence of C4d deposits has been classified as a new Banff parameter for ABMR [9].

In order to analyze antibody-mediated rejection without C4d deposits, in this work we will present the clinical case of a patient who presented an ABMR episode 4 days after kidney transplantation.

## 2. Materials and Methods

### 2.1. HLA Typing

Intermediate-resolution genotyping of HLA class I (A, B and C) and II (DRB1, DQB1) loci was performed using the LIFECODES HLA-SSO Typing kits sequence-specific oligonucleotide test (IMMUCOR GTI Diagnostics, Inc., 20925 Crossroads Circle. Waukesha, WI 53186 USA). LIFECODES HLA-SSO Typing kits utilize sequence-specific oligonucleotides (SSOs) to identify which HLA alleles are present in a PCR amplified sample. Target DNA was amplified by PCR using sequence-specific primers, followed by hybridization with allele-specific oligode-oxynucleotides coupled with fluorescent phycoerythrin-labelled microspheres. Fluorescence intensity was determined using a LABScan 100 system (Luminex xMAP, Austin, Texas, USA). HLA alleles were assigned using the Match It DNA software (IMMUCOR).

### 2.2. LABScreen™ Single Antigen HLA Class I and Class II

The identification of HLA-antibodies was performed using LABScreen kits (One Lambda, Canoga Park, California, USA). This test uses microbeads coated with purified class I or class II HLA antigens and pre-optimized reagents for the detection of class I or class II HLA antibodies in human sera. The Single Antigen assay allows confirmation of antibody specificity. The individual single beads are used to focus on reactions against one or various antigens. Fluorescence intensity was determined using a LABScan™ 100 (Luminex^®^ 100/200) for analysis of up to 100 or 500 bead regions, respectively, in a single test. The cutoff was set at 1500 MFI for class I and class II alleles [18].

### 2.3. C1qScreen™ Single Antigen HLA Class I and Class II

Determination of complement binding capacity was performed using the C1qScreen™ (One Lambda, Canoga Park, California, USA). C1qScreen is used for the detection of complement-binding anti-HLA antibodies in serum samples. One Lambda’s HLA LABScreen™ and C1qScreen™ Assays allow detection and identification of IgG antibodies directed against complement-fixing HLA class I and class II molecules. The method is based on the specificity of the antigen-antibody binding (Ag-Ab), recognized by the C1q molecule. The HLA LABScreen™ and C1qScreen™ beads are incubated with the serum sample to be tested. Each sphere contains a single antigen of HLA. The anti-HLA antibodies present in the sample bind specifically to each sphere, which allows the identification of the antigens against which the patient is sensitized. The addition of the C1q molecule recognizes the Ag-Ac complex and binds exclusively to the Fc portion of the anti-HLA antibodies that fixa C1q and are capable of activating the complement cascade. Finally, an anti-human C1q antibody labeled with Phycoerythrin (PE) is used, which serves as an indicator of the presence of complement-fixing antibodies by emitting a fluorescence signal. Fluorescence intensity was determined using a LABScan™ 100 (Luminex^®^ 100/200) for analysis of up to 100 or 500 bead regions, respectively, in a single test. The cutoff was set at 1500 MFI for class I and class II alleles.

## 3. Results

A 54-year-old male patient developed end-stage renal failure due to nephroangiosclerosis in 2016. After diagnosis, the patient began hemodialysis treatment in April 2016. Personal history: Type 2 diabetes mellitus treated with insulin (pretransplantation), arterial hypertension, duodenal ulcer, ex-smoker (20 cigarettes per day), occasional drinker, syncope episodes at the start of hemodialysis attributed to treatment with doxazosin.

For the inclusion of the patient on the kidney transplant waiting list, the presence of anti-HLA antibodies (anti-HLA Ab) was determined (Table 1 and Figure 1) and the HLA typing performed (Table 2).

After 1 and a half years on the waiting list, on 30 January 2018, the patient was transplanted with a kidney from a deceased donor. HLA compatibility between donor and recipient A: 1, B: 0, C: 0, DRB1: 0, DQB1: 0 (they shared the HLA-A*30 antigen). Additionally, it was observed in the HLA typing that the donor is homozygous for the public antigen Bw4 and the recipient is homozygous for the public antigen Bw6 (Table 2).

At the time of transplantation, a final complement-dependent cytotoxicity (CDC) crossmatch was performed with a negative result for T and B lymphocytes, as well as a negative virtual crossmatch (absence of DSA by the Single Antigens Beads (SAB) technique) (Figure 1).

The first 3 days after transplantation, the patient improved his serum creatinine levels (levels on the day of transplantation were 5.46 mg/dL), not requiring dialysis (Table 3). This showed a rapid adaptation and response of the graft, being able to discard the presence of delayed graft function (DGF) [19]. Likewise, diuresis and blood pressure remained at normal levels (Table 3).

On the fourth post-transplant day, the patient presented delayed kidney graft function with elevated serum creatinine levels (SCR) (SCR: 5.31 mg/dL). It was decided to determine anti-HLA antibodies by SAB to identify the presence of possible DSA, finding positivity against the public antigen Bw4 (Figure 2). The antibody identification result shows a positive reaction against all antigens that share the Bw4 epitope. The determination of antibodies against MICA (major histocompatibility complex class I chain-related gene A) was also carried out, the result being negative.

In accordance with the observed results, it was studied whether the antibodies present post-transplantation were complement fixers, using the One Lambda C1q technique. The results obtained were negative, showing that the antibodies present are not complement-fixing antibodies (Figure 3).

The results of the renal graft biopsy showed the presence of lymphocytic infiltrate, as well as the presence of C4d deposits in the glomeruli but absence of C4d in the peritubular capillaries (Figure 4 and Figure 5).

Subsequently, the pretransplant sera was again analyzed for the presence of DSA, lowering the cutoff Single Antigen to 500 MFI. The results in all the sera were also negative. In the serum of the day 10 January 2018, Cw12 showed an MFI value of 638, which is an antigen from the donor. However, in that same determination we had an MFI of 538 for Cw2, a self antigen. Therefore, we could not give value to these antibodies. Likewise, we reanalyzed the antibody pattern on 2 June 2018 (Figure 2), observing that the MFI for Cw12 drops to 70, demonstrating that there was no reaction against this DSA.

The results obtained confirmed ABMR rejection of the kidney graft in the presence of non-complement-fixing antibodies. Rescue immunosuppressive treatment was started with methylprednisone, thymoglobulin, plasmapheresis, and immunoglobulins, who presented a complete response and recovery of renal function. After the rescue treatment, the determination of antibodies by SAB was carried out again, finding that the DSA had disappeared.

The patient suffered other post-transplant complications such a wound infection with pseudomonas and cytomegalovirus infection, which were resolved with antibiotic treatment, without compromising kidney function.

As of October 2020, 2 years and 9 months after transplantation, the patient had stable renal function with SCR of 1.22 mg/dL. Control ultrasound results showed native kidneys decreased in size, increased cortical echogenicity without defined solid nodules, or dilated excretory system. The kidney graft located in the right iliac fossa measured 10.5 cm in size, with preserved morphology and echostructure, without solid nodules or dilatation of its collecting systems. The bladder was distended without significant alterations.

The results of the Doppler examination showed a permeable renal artery and vein, persisting the focal increase in anastomosis velocity of up to 200 cm/s, without intrarenal repercussion. The resistance index in intrarenal arteries was within normality. At the evaluation, the iliac vessels did not show any significant alterations.

The DSA antibody identification study, carried out on 26 March 2021, showed negative results for class I and class II.

Currently, the patient continues with immunosuppressive treatment, consisting of doses of Mycophenolate Mofetil, Prednisone, Omeprasol, Magnesium, Candesartan, Rosuvastatin, Hydroferol, Linagliptin, Tacrolimus, Carvedilol, Paricalcitol and Manidipine.

## 4. Discussion

An important point to consider in organ transplantation is the compatibility of the HLA molecules between donor and recipient. However, due to their high polymorphism and their frequency in the population, finding a donor with 100% compatibility represents a challenge for the different transplant groups. On the other hand, the frequency of the public epitopes Bw4 and Bw6 in the population, estimated between 73% and 86% respectively [8], results in a high rate of sensitization to any of these epitopes, when the receptor has a sensitizing event. Currently, HLA compatibility is part of the solid organ allocation policies, evaluating at least the compatibility at the A, B and DR loci, the loci that represent the majority of HLA polymorphisms. However, compatibility in public antigens is not part of the criteria for organ allocation [20].

After a sensitizing event that has had a receptor, the resulting antibodies directed against HLA molecules can be specific and react against private antigens or react against public antigens. Anti-HLA antibodies have the ability to recognize foreign proteins that differ by only one amino acid and, in the same way, can cross-react between different HLA [7]. These antibodies are strongly related to ABMR and decreased graft survival. On the other hand, the specificities of antibodies formed by the recognition of the public antigens Bw4 and Bw6 by allogeneic T cells demonstrate that these antigens play an important role in graft survival, especially when the donor/recipient is homozygous for them [21]. We know then that the public antigens Bw4 and Bw6 are expressed in the HLA-B alleles and in some HLA-A alleles, therefore, the incompatibility of these molecules can represent a high load of epitopes different from those present in the receptor [22].

Our clinical case study showed that ABMR developed in just 4 days after transplantation, with no evidence of pre-transplant DSA. This patient was incompatible with the donor in the public antigens Bw4/Bw6, being the recipient homozygous for the public antigen Bw6 and the donor homozygous for Bw4. Evidence shows that the presence of preformed antibodies in the recipient, directed against some HLA-B antigens that share the public Bw4 epitope, led to preformed antibodies against the donor’s cross-reactive group (CREG). The biopsy showed ABMR, described according to Banff criteria and without C4d deposits. Our results are similar to those described by Nainani et al. [23], who showed that antibodies can react against the different CREG groups present in the donor, increasing the risk of early ABMR.

On the other hand, the possibility that the recipient had specific memory reactive cells against the donor’s HLA antigens is discussed, which in the face of the antigenic stimulus of the transplant, would induce the novo production of DSA antibodies, which were not identified in the determination of anti-HLA antibodies done to the patient during his time on the waiting list. As discussed by Gorbacheva et al. [24], there is evidence that donor-specific memory T cells, for which there is no routine test that allows their identification, are potent inducers of early ABMR episodes in recipients of kidney transplant.

Another important finding in our clinical case is the evidence that the DSA against the public antigen Bw4 observed post-transplantation showed mean fluorescence intensity (MFI) values of less than 4000. This MFI is associated with the complement fixation capacity of DSA antibodies, which is consistent with previous studies [24,25]. These suggest that MFI values lower than 5000 reflect a low antibody titer and correlate with a poor complement fixation capacity and, despite the fact that the patient had an ABMR rejection event, there was no loss of the kidney graft.

The importance of early post-transplant monitoring of DSA is highlighted, which is reflected in the possibility of a timely intervention thus avoiding graft loss. In addition, the classification of these antibodies, according to their complement fixation capacity, could be a valuable tool in post-transplant monitoring, together with determining the presence of DSA before performing a kidney biopsy [17,26].

## 5. Conclusions

The ABMR evidence in our clinical case study demonstrates the importance of considering the compatibility of public antigens in the assignment of kidney transplantation, especially in those homozygous patients who are sensitized against alleles associated with a public specificity and who could receive a double antigenic load. On the other hand, graft survival in the patient in our study, despite the presence of DSA, confirms the predictive factor of determining the complement fixation capacity of DSA antibodies. This allowed risk stratification and early intervention of the patient with rescue therapy, which led to the recovery of renal function, as well as the disappearance of DSA antibodies, which suggests that its determination could be useful in the post-transplant monitoring of the kidney patient.

## Figures and Tables

**Figure 1 antibodies-10-00028-f001:**
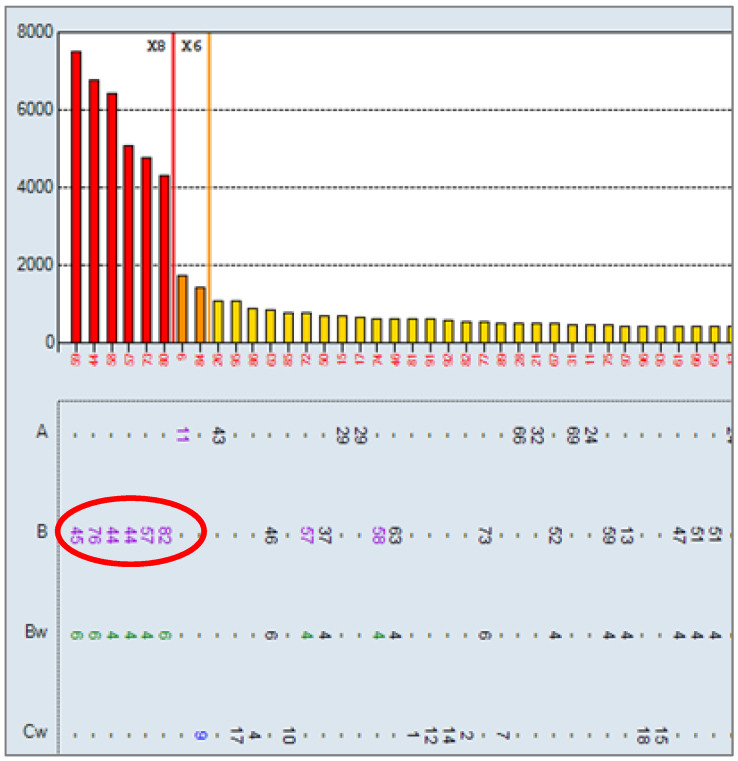
Single Antigen Class I results at 10 January 2018: positive for the specificities B45, B76, B44, B57, B82, A11, Cw9. HLA class II negative. The cutoff for positivity was set at 1500 MFI.

**Figure 2 antibodies-10-00028-f002:**
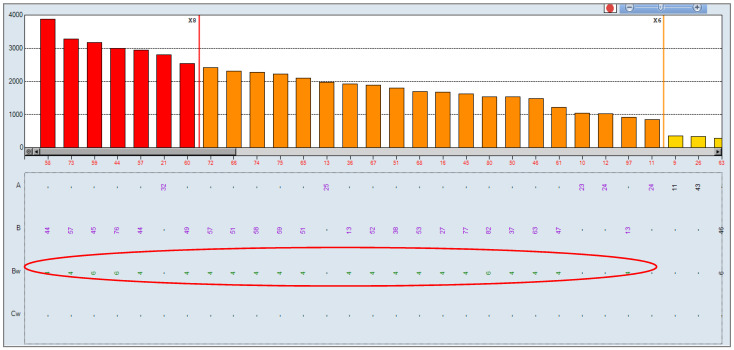
Single Antigen Class I at 6 February 2018: positive for all Bw4 specificities. The cutoff for positivity was set at 1500 MFI.

**Figure 3 antibodies-10-00028-f003:**
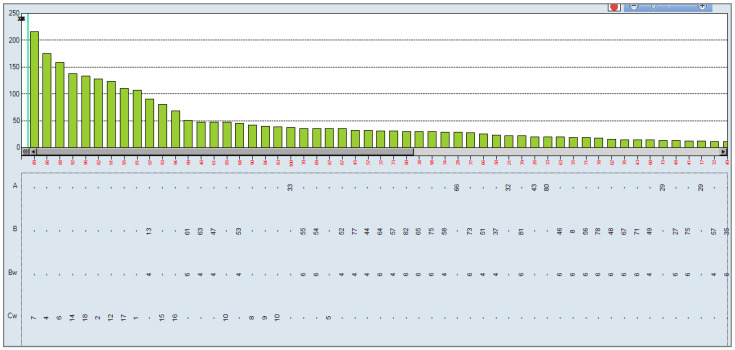
C1q assay: negative for all specificities. The cutoff for positivity was set at 1500 MFI.

**Figure 4 antibodies-10-00028-f004:**
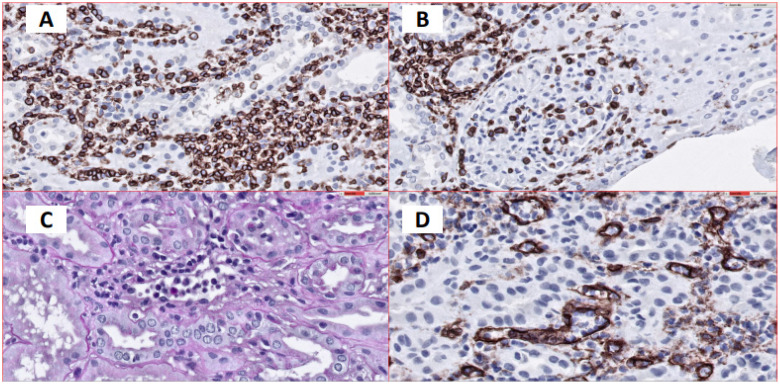
(**A**). Interstitial and tubular inflammatory cellular infiltrates. Immunohistochemical staining for CD45 expression, 40×. (**B**). Evidence of inflammatory cells in glomeruli (Glomerulonephritis). Immunohistochemical staining for CD45 expression, 40×. (**C**). Dilated peritubular capillaries infiltrated by inflammatory cells. PAS stain, 60×. (**D**). Dilated peritubular capillaries visualized by CD34 immunohistochemical staining 60×, which shows endothelial cells. Peritubular capillaries are infiltrated by inflammatory cells.

**Figure 5 antibodies-10-00028-f005:**
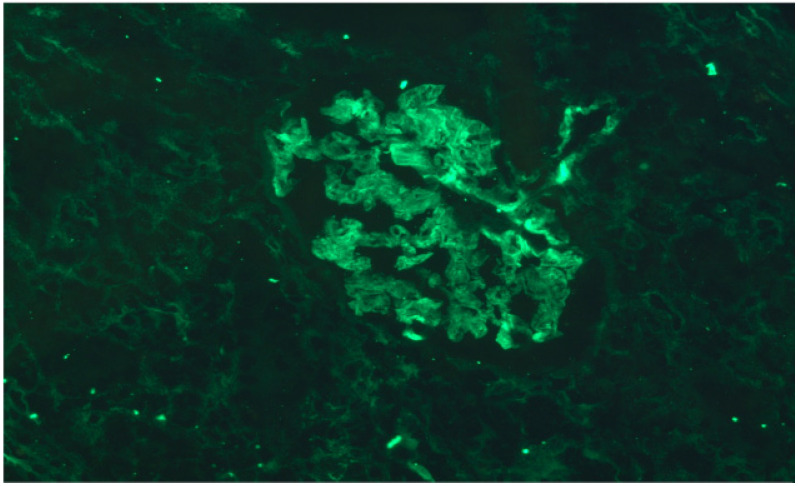
No C4d deposits were found in peritubular capillaries, but there is evidence of antibody interaction with the glomerular epithelium. Indirect immunofluorescence 40×.

**Table 1 antibodies-10-00028-t001:** Results of the identification of anti-HLA antibodies against class I and II, carried out on the patient during the time on the waiting list for kidney transplantation.

Anti-HLA Antibodies Analysis Date	Anti-HLA CLASS I Antibodies	Anti-HLA CLASS II Antibodies
14 March 2017	B45, B44, B76, B82	Negative
6 April 2017	Negative	Negative
7 July 2017	Negative	Negative
9 October 2017	Negative	Negative
10 January 2018	B45, B44, B76, B57, B82, A11, Cw9	Negative

**Table 2 antibodies-10-00028-t002:** Recipient and donor HLA typing. Compatibility between recipient and donor 1 HLA allele out of 10. Mismatch in specificity Bw is highlighted in red (* Molecular typing).

	A *	A *	B *	B *	Bw	Bw	C *	C *	DRB1 *	DRB1 *	DQB1 *	DQB1 *
HLA recipient	01	30	18	40	Bw6	-	02	06	11	-	03	-
HLA donor	03	30	38	-	Bw4	-	12	-	01	13	05	06

**Table 3 antibodies-10-00028-t003:** Graft evolution markers during the first 3 days after transplantation. cc: cubic centimeters; mmHg: millimeters of mercury.

	Creatinine	Diuresis	Blood Pressure
Day 1 post-transplant	3.44 mg/dL	3480 cc	137/94 mmHg
Day 2 post-transplant	2.46 mg/dL	5260 cc	141/92 mmHg
Day 3 post-transplant	2.28 mg/dL	8910 cc	145/90 mmHg

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
