# Peer review of "Clinical Case: Patient with Mixed Graft Rejection Four Days after Kidney Transplantation Developed Specific Antibodies against Donor Bw4 Specificities"

_2073-4468, 2021, doi:10.3390/antib10030028_

Round 1

Reviewer 1 Report

The manuscript "Clinical case: Patient with mixed graft rejection four days after kidney transplantation developed specific antibodies against donor Bw4 specificities" describes a kidney transplant case with post-transplantation renal dysfunction likely due antibodies with Bw4 specificities.

 They provide evidence to support that CREG antibodies are responsive for the rejection in the case. The data presented are detailed and adds another piece of evidence that antibody against public antigens are important in kidney rejection.

Fig.4 legend: …D. Dilated peritubular capillaries visualized by CD34 (?).....  CD34 is a marker for hematopoietic stem cell not for inflammatory cells

Page 2 Line 54  T cell-mediated rejection (TCMR), (ABMR), and ... --> remove the ( )

Page 7 Line 184.  the patient has stable renal function with CSR...  --> SCR

Page 7 line 214 and 222 --> use ABMR instead of the antibody-mediated rejection (ABMR)  à stick to the abbreviation usage rule

Page 8, Line 228... the donor's CREG cross-reactive group... àThe CREG is redundant.

Figure 1. The meaning of the dark circle in this figure is unclear. It seems to miss target and lack explanation.

Page 7  Line 183.  In this paragraph, the authors use present tense to describe the post-transplantation situations.  

Page 7, Line 192. "Resistance index in intrarenal arteries within normality. Iliac vessels results without significant alterations." ---> not complete sentences.

Reference 19 has some irrelevant Chinese words and lacks the complete author list and title of the reference

Author Response

We would like to thank the Referee for these important comments.

We agree with you that CD34 is a marker for hematopoietic stem cells. In addition, it is also a marker of vascular endothelial cells and is used to mark said structures.

Fina L, Molgaard HV, Robertson D, Bradley NJ, Monaghan P, Delia D, Sutherland DR, Baker MA, Greaves MF. Expression of the CD34 gene in vascular endothelial cells. Blood. 1990 Jun 15;75(12):2417-26. PMID: 1693532.

We have made the suggested changes and we hope that the new version of the manuscript will be ready for publication in Antibodies.

Again, thank you very much again for your valuable comments.

Reviewer 2 Report

the case report written by C. Munoz-Herrera presents the case of a transplanted patient with mixed allograft rejection caused by dnDSA against Bw4 4 days after renal transplantation. Although the authors did already a precise work-up some relevant informations are still missing:

  • what was the pre-/posttransplant MFI cut-off classifying a detected SAB as positive (e.g. MFI > 500?);
  • as illustrated in figure 1 the recipient was positive in 01-2018 also for Cw4, which might be relevant as B44 is not only part of the „Bw4-family“, but is also closely associated with the Cw4 locus; have donor and recipient been characterized regarding Cw4 or Cw6 serotypes (B45/B57 closely associated with Cw6); may be that pre-existing Cw-DSA - of course only if verifiable – triggered/enhanced mixed allograft rejection;
  • in figure 3 the authors state that C1q assay was negative for all specificities; nevertheless apart from Cw7 Cw4 and Cw6 showed the most pronounced peaks; so precise characterization on donor/recipient Cw4 and Cw6 compatibility needs to be done;
  • the authors should add some more informations on the clinical course of the pts. within the first 3 days after transplantation, e.g. CIT? DGF? diuresis? arterial hypertension? Have the authors checked for non-HLA antibodies (e.g. AT1R)? which might also mimic AMR and which might be linked with hypertension after transplantation (nephroangiosclerosis recipients renal disease)

In summary, some additional work should be done, to be sure, that development of de-novo Bw4 DSA took place, without any recipient pre-immunization by donor antigens really within 4 days, which would represent a very intensified kinetics for IgG antibody formation.

Author Response

We thank the Referre for these important reflections on our work.

  • what was the pre-/posttransplant MFI cut-off classifying a detected SAB as positive (e.g. MFI > 500?);

We have introduced the cut-off used for positivity in the determination of DSA and the C1q test (MFI > 1500).

  • as illustrated in figure 1 the recipient was positive in 01-2018 also for Cw4, which might be relevant as B44 is not only part of the „Bw4-family“, but is also closely associated with the Cw4 locus; have donor and recipient been characterized regarding Cw4 or Cw6 serotypes (B45/B57 closely associated with Cw6); may be that pre-existing Cw-DSA - of course only if verifiable – triggered/enhanced mixed allograft rejection

In figure 1, the circle marking positive DSAs is wrong. It has been corrected in the new version of the manuscript and we apologize for this mistake. Furthermore, the MFI cutoff is at 1500, thus there is no presence of DSA against the HLA-C*04 allele, i.e., the recipient is not positive for Cw4 antibodies.

Likewise, Table 2 shows the complete typing of donor and receptor (HLA-A, B, C, DRB1 and DQB1) and there is no presence of incompatibility at the HLA-C locus, since the donor is homozygous for the HLA-C*12 allele and there were no antibodies against that allele.

  • in figure 3 the authors state that C1q assay was negative for all specificities; nevertheless apart from Cw7 Cw4 and Cw6 showed the most pronounced peaks; so precise characterization on donor/recipient Cw4 and Cw6 compatibility needs to be done

We have introduced the cutoff for the C1q test (MFI >1500) showing that the result is negative. As well, although the highest peaks are against the HLA-C*07, C*06 and C*04 alleles, they are not alleles present in the donor.

  • the authors should add some more informations on the clinical course of the pts. within the first 3 days after transplantation, e.g. CIT? DGF? diuresis? arterial hypertension? Have the authors checked for non-HLA antibodies (e.g. AT1R)? which might also mimic AMR and which might be linked with hypertension after transplantation (nephroangiosclerosis recipients renal disease)

We have recovered some of the clinical data that you suggested during the first three days after transplantation (Table 3). Thus, being able to rule out the presence of DGF.

We have not performed the determination of antibodies against AT1R, but the absence of hypertension (Table 3) can help us to discard nephroangiosclerosis recipients renal disease.

We also determined anti-MICA antibodies with a negative result.

We hope that the new version of the manuscript will be ready for publication in Antibodies.

Again, thank you very much for your valuable comments.

Round 2

Reviewer 2 Report

the authors gave some additional clarifications, which improved quality of the paper. However, although from a clinicians perspective reasonable to choose a MFI cut-off of 1500, in this specific situation a re-analysis of preTx DSA expression with a lower MFI cut-off for positivity (e.g. 500) would be worth, to exclude any impact of pre-existing low-level DSA, which when - if detectable - caused/triggered (very early) AMR. At least the latest sera samples before kidney transplantation should be re-analysed.

Author Response

We thank the Referee for this important comment.

It is very interesting to reanalyze the sera with the aim of finding a possible DSA.

We have retested the pre-transplant sera, the results being negative.

We have added these results to the manuscript in the Results section on page 8, line 202.

We hope that the new version is ready to be published in Antibodies.

This manuscript is a resubmission of an earlier submission. The following is a list of the peer review reports and author responses from that submission.